# “Immuni” and the National Health System: Lessons Learnt from the COVID-19 Digital Contact Tracing in Italy

**DOI:** 10.3390/ijerph19127529

**Published:** 2022-06-20

**Authors:** Silvia Ussai, Marco Pistis, Eduardo Missoni, Beatrice Formenti, Benedetta Armocida, Tatiana Pedrazzi, Francesco Castelli, Lorenzo Monasta, Baldassare Lauria, Ilaria Mariani

**Affiliations:** 1Clinical Pharmacology and Toxicology, University of Cagliari, 09124 Cagliari, Italy; 2Section of Neuroscience and Clinical Pharmacology, Department of Biomedical Sciences, University of Cagliari, 09124 Cagliari, Italy; mpistis@unica.it; 3Center for Research on Health and Social Care Management (CERGAS), Bocconi University, 20124 Milan, Italy; eduardo.missoni@unibocconi.it; 4Saluteglobale.it Associazione di Promozione Sociale, 25123 Brescia, Italy; 5Department of Infectious and Tropical Diseases, University of Brescia and ASST Spedali Civili of Brescia, 25123 Brescia, Italy; beatriceformenti1992@gmail.com (B.F.); francesco.castelli@unibs.it (F.C.); 6Istituto Superiore di Sanità, Department of Cardiovascular, Endocrine-Metabolic Diseases and Ageing, 00161 Rome, Italy; benedetta.armocida@iss.it; 7Department of Occupational Medicine, Hygiene, Toxicology and Occupational Prevention, University of Brescia and ASST Spedali Civili of Brescia, 25123 Brescia, Italy; tatiana.pedrazzi@gmail.com; 8Clinical Epidemiology and Public Health Research Unit, Institute for Maternal and Child Health—IRCCS “Burlo Garofolo”, Trieste, Italy; lorenzo.monasta@burlo.trieste.it; 9G. Gulotta” Foundation, 50100 Florence, Italy; sasalauria@yahoo.it; 10WHO Collaborating Centre, Institute for Maternal and Child Health—IRCCS “Burlo Garofolo”, 34137 Trieste, Italy; ilaria.mariani@burlo.trieste.it

**Keywords:** immuni, Italy, contact tracing system, COVID-19

## Abstract

Since the early stage of the current pandemic, digital contact tracing (DCT) through mobile phone apps, called “Immuni”, has been introduced to complement manual contact tracing in Italy. Until 31 December 2021, Immuni identified 44,880 COVID-19 cases, which corresponds to less than 1% of total COVID-19 cases reported in Italy in the same period (5,886,411). Overall, Immuni generated 143,956 notifications. Although the initial download of the Immuni app represented an early interest in the new tool, Immuni has had little adoption across the Italian population, and the recent increase in its download is likely to be related to the mandatory Green Pass certification for conducting most daily activities that can be obtained via the application. Therefore, Immuni failed as a support tool for the contact tracing system. Other European experiences seem to show similar limitations in the use of DTC, leaving open questions about its effectiveness, although in theory, contact tracing could allow useful means of “proximity tracking”.

## 1. Introduction

Since the early stage of the current pandemic, digital contact tracing (DCT) has been considered globally as a complement to control interventions [1,2].

On 17 March 2020, the Italian Prime Minister established urgent measures to improve the response to the COVID-19 public health emergency. This included, among others, the introduction of DCT systems to determine the extent of the outbreak and early detection of COVID-19 exposure among the population [3]. As such, the Ministries of Health and Economic Development jointly launched a fast call for contributions on COVID-19 tracing software applications (319 proposals received) [4]. The solution selected in Italy, called “Immuni”, is a DCT built on a circular risk model: whenever two smartphones with the app installed are within a 2 m range of each other for a period longer than 15 min, they automatically exchange codes generated by the app, enabling contact tracing. When the Local Health Authorities (LHAs) register a COVID-19 case, the certified information is uploaded to the Immuni server; any available epidemiological information from the previous 14 days is uploaded too. Subsequently, the app notifies the proximity risk to the end user, which is expected to refer to the LHA for the appropriate case management. Immuni technology is based on rolling proximity identification via Bluetooth, a high-level cryptography method to preserve user’s privacy, and the data collection is exclusively managed by a public agency [5].

Immuni has been downloadable on a voluntary basis since 1st June 2020, with a seven day trial from 8 June 2020 limited to four Regions (Liguria, Marche, Molise, Puglia). During this period, the developers fixed some technical issues; however, no co-creation activity with the end-users has been conducted to explore the acceptability (willingness to engage with the intervention), adaptation (likelihood to have value and impact for the population), adherence (the degree to which the user follow the intervention as it was designed) and compliance (the consistency and accuracy with which users follow the intervention) of the app [6]. 

Additionally, public initiatives to stimulate the uptake of Immuni failed in enhancing its perceived benefits and self-efficacy. Indeed, while Italy has just exited it’s fourth wave of COVID-19 (reaching more than 200,000 daily positive tests in January 2022), a national survey revealed that almost 40% of the citizens are still reluctant to use Immuni [7], due to (i) privacy concerns on medical records, as well as the risk of false positives leading to unnecessary quarantine; (ii) technical barriers: for instance, the app cannot be installed on every type of phone, and the population at the highest risk—the elderly—are unlikely to use smartphones; (iii) the lack of trust in government technology, particularly after the evidence that the LHAs (responsible for the delivery of care) did not integrate Immuni within their information systems, leading to an interruption of the data cycle. The importance to explore the health belief model as proposed by Rosenstock has been also reflected at European level. In 2020, Belgium launched a survey—administered to 1500 respondents, aged 18 to 64 years of age, showing that approximately 50% of respondents declined to use a COVID-19 tracing app. The most important predictor was the perceived benefits of the app, followed by self-efficacy and perceived barriers [8]. 

Since July 2021, Immuni is also one of the two possible modalities to download and store the Green Pass certification, the EU digital COVID certificate released to vaccinated people, those who have recovered from the infection, and those with a negative COVID-19 test, which is currently required to access most of the public offices and public transport, and is also mandatory to access many workplaces [9].

This opinion article aims to describe the use of Immuni in Italy from 1 June 2020 to 31 December 2021 and investigate the app utilization during the pandemic response.

## 2. Materials and Methods

Data about Immuni are provided by the Minister of Health [9] and retrieved from the official Immuni platform. The variables examined were the daily numbers of Immuni downloads for both iOS and Android, the daily number of notifications sent by the app, and the daily number of users tested positive to SARS-CoV-2 who uploaded their status to the app. The time trend of the notifications sent by Immuni was assessed using a Mann–Kendall test. Spearman’s rank correlation was adopted to investigate the correlation between Immuni variables, and to compare Immuni positive users with the daily numbers of SARS-CoV-2 positive cases provided by the Minister of Health in the same period, defined as the sum of hospitalized patients and in self-isolation. The trend of total positive cases, Immuni downloads, notifications sent, and Immuni positive users was represented in the form of daily absolute numbers or percentages, beginning on 15 June 2020, the date when the first positive user was recorded by Immuni.

## 3. Results

### Description of Results

By the 31 December 2021, the cumulative number of Immuni app downloads had reached 19,151,200 (36.7% of the eligible population, >14 years old), and Immuni had identified 44,880 COVID-19 cases, which corresponds to less than 1% of total COVID-19 cases reported in Italy in the same period (5,886,411). Overall, Immuni generated 143,956 notifications, most of which were sent in October and November 2020 (Figure 1a). The peak was reached on the 27 October 2020, with 5195 notifications for the 143 users who tested positive (0.0015%). 

Excluding these two months, the notification trend is flat (trend test *p* = 0.133) with a slight but not statistically significant increase from December 2021.

The number of users who tested positive was moderately correlated with the daily positive cases (rho = 0.68, *p* < 0.001) and constantly below 300 cases, even during the second and third COVID-19 waves in Italy (December 2020 and April 2021) when total positive cases exceeded 800,000 and 550,000 cases, respectively. From 15 December 2021, in correspondence with the beginning of the forth COVID-19 wave, the number of positive users increased to over 300 cases per day and reached 1652 cases on 31 December 2021, yet this increase never exceeding 0.01% of the total users (Figure 1b).The correlation between Immuni download and notifications sent to positive users was negligible (rho = −0.18 and 0.14 respectively, *p* < 0.001) indicating an extremely low uptake of Immuni, thus a failure to significantly contribute to contact identification during the epidemic.

## 4. Implementation in Clinical Practice

As contact tracing remains a crucial component of the COVID-19 response, DCT should be reviewed and enhanced to be effective in Italy, particularly considering the challenges revealed during the rapid spread of the Omicron variant, which has put the traditional contact tracing system under extreme pressure. As manual tracing operations take about 12 h and require, on average, three units of specialized personnel, the public health system may become burdened, both in terms of human resources and physical resources (scarcity of swabs and reagents etc.), leaving laboratories overloaded [10]. Ferretti et al. have shown that up to 55% of transmissions come from asymptomatic carriers, and the virus has a very short generation period (3–5 days on average). This means that the traditional transmission chain monitoring systems, essentially based on diagnostic test results, may not be rapid enough to limit the spread of COVID-19 and, therefore, to interrupt its epidemic reproduction. Furthermore, COVID-19 positive patients may find it difficult to remember all their previous contacts from 14 days prior to the test [11]. Several contacts may have also occurred accidentally (in the supermarket, on public transportation, at the post office etc.) and involving strangers, resulting in a lack of direct knowledge as to the contacts’ identities. Since about 44% of secondary cases of infection occur during the pre-symptomatic phase of primary cases, the role of digital tracing combined with social distancing and isolation may be important to improve the accuracy of the process, with the understanding that the contact tracing app alone is insufficient to limit infections (e.g., it will not protect those who do not own a smartphone or who, for various reasons, have not installed the app). Hence, apps need to be relied on in addition to diagnostic testing and interventions of healthcare professionals [12].

The pressure the Italian public health system experienced amid the spread of omicron also had a direct impact on the tracking system, and these pressures forced the Government to make some widely criticized decisions regarding different quarantine rules for fully vaccinated and non-vaccinated populations who encountered COVID-19 positive cases, and calling for the army to support the healthcare facilities in their daily contact tracing activities. In light of this, the proper and widespread use of technology for contact tracing could allow for a more efficient means of “proximity tracking”, because the tool can inform operators who may intervene quickly by taking preventative action and decide, supported by evidence, which contacts are considered “at risk” and which are not. Among other benefits, this approach will allow for a massive reduction of unnecessary quarantines (in January 2022, 3 million Italians were self-isolated) [13].

A further element to be considered in clinical practice is related to the app’s interoperability. The European Union (EU) recommends that Member States identify and implement a common strategy, including the establishment of guidelines for the development of contact tracing apps that work across the Union. In April 2020, the European Commission presented the EU’s toolbox on contact tracing and warning apps, and their interoperability gateway, ensuring that Member States’ apps could work seamlessly cross-borders. However, to date, the number of countries that fully adopted the gateway (countries registered with the interoperability gateway service) remains limited [14].

Additionally, there are several options in terms of data storage in EU countries. These are: (i) the use of a centralized system, in which all data are stored in a single central server; or (ii) the use of a decentralized system, in which the data are collected via individual devices and are only transmitted in the event that a person who has been using the app should test positive for COVID-19 (this option includes the need to set up motion alerts). Germany, Italy, Switzerland, and Austria opted for a decentralized system, while Belgium adopted the so called minimization principle (storing as little data as possible in servers, and only transmitting data as warnings in cases of risky contacts) [15].

States are also obliged to indicate the purposes of the app with utmost clarity; the data must be managed by government bodies, namely health authorities.

## 5. Discussion

Overall, although the initial download of the Immuni app reflected the initial interest in the new tool, it has had little adoption across the Italian population, and the recent increase in its download rate is likely to represent the extraction of the EU COVID digital certificate (“Green Pass”), which is mandatory in Italy and can be obtained through the app [16]. 

The evidence of low uptake also relates to the high fragmentation in the Italian National Health Service (Servizio Sanitario Nazionale, SSN), which is a regionalized and semi-federalized system. Indeed, some politicians and Regions expressed several concerns about the app, creating confusion and undermining the population’s willingness to accept the app, raising questions regarding the referral guidelines across healthcare providers. Furthermore, in parallel with the Government, Sardinia (1.7 million residents) and Sicily (5 million residents) developed their own contact tracing apps (namely, Sardegna Sicura and Sicilia SiCura) [17]. 

In addition, several technical issues limiting Immuni’s contribution to the contact tracing system are related to the unknown number of real users (as the app can be downloaded by the same person using multiple smartphones; or can just be downloaded to obtain the “Green Pass”), the inconsistent evaluation of other variables for determining the risk (i.e., the use of masks or if contact happened in the open air), and the fact that the app does not work if the Bluetooth is inactive. Without a solid model including these factors, the app appears to identify just phones rather than epidemiological risks [18,19].

The introduction of digital innovations requires relevant steps addressing technical, political-strategic, and human challenges. This includes, among others, the design of context-specific and population-based controlled pilots before the final deployment of the tool. In this light, an adaptation of the Technology Acceptance Model (TAM) proposed by Davis in 1989 may be investigated as an influential model of technology acceptance, with two primary factors influencing individual’s intention to use new technology: perceived ease of use and perceived usefulness [8]. 

Similar findings on the relevance of acceptance models while introducing a DCT system have been described [20] amid a 4 week trial held in San Sebastián de la Gomera, in the Canary Islands, during July 2020. On this occasion, researchers were successful in simulate four outbreaks of COVID-19 among the 10,000 inhabitants, tracked their behavior, and then computed 7 key performance indicators to evaluate the Spanish “Radar COVID” app’s effectiveness (these indicators included: “adoption”, “adherence” (i.e., whether or not the app was used 10 days after download), “compliance” (whether or not codes were entered into the app), and overall detection rates (“the average number of close-contacts of a given infected individual which are notified by the app”).

Looking at the international context regarding the DCT systems in use during the pandemic, Wymant et al. found that the British app was downloaded on 21 million separate devices and regularly used in the target period by 49% of the eligible population. Overall, approximately one new case was averted for each individual confirmed to have COVID-19 who consented to their contacts being notified through the app [21].

In Switzerland, the SwissCovid app was downloaded more than 3.1 million times (36% of eligible population) in one year from its launch, and has approximately 1.7 million active users [22]. By comparison, the German Corona-Warn-App was downloaded by 34% of the eligible population and sent exposure notifications following 478,000 positive test results. The associated survey conducted across the target population also suggest suggests a pessimistic view on the effectiveness of app-based contact tracing to contain the COVID-19 pandemic in Germany [23]. In the Netherlands, there were 4.96 million app downloads (28.7% of the total population of 17.3 million) and 178,000 issued exposure notifications. In this country, a recent usability study indicates that the CoronaMelder app is easy to use, but participants experienced misunderstandings about its functioning e.g., perceived lack of clarity led to misconceptions about the app, mostly regarding its usefulness and privacy-preserving mechanisms [24].

## 6. Conclusions

Other European experiences seem to show limitations in the use of DTC, similar to those experiences Italy has had with Immuni [25] during the pandemic response, leaving open questions about its implementation in the real-world. As traditional contact tracing remains a crucial component of the COVID-19 response, mobile apps can offer promise, especially when considering the speed and scale required for tracing to be effective. However, the app should be presented as part of the national coronavirus measures instead of as a stand-alone app offered to the public, particularly when the health system and services are fragmented [26]. 

The uptake of digital contact tracing apps may be affected by several perceived risks (e.g., privacy) that are not compensated by the potential benefits, especially if the benefits are not clearly presented to the population. Authorities have requested high levels of transparency during the setting up of DCT tools, including a proactive communication addressing ethical, legal, and social issues related to such technologies prior to their launch.

Further research on the DCT development, roll-out and evaluation is needed, especially to identify context-specific functionalities.

## Figures and Tables

**Figure 1 ijerph-19-07529-f001:**
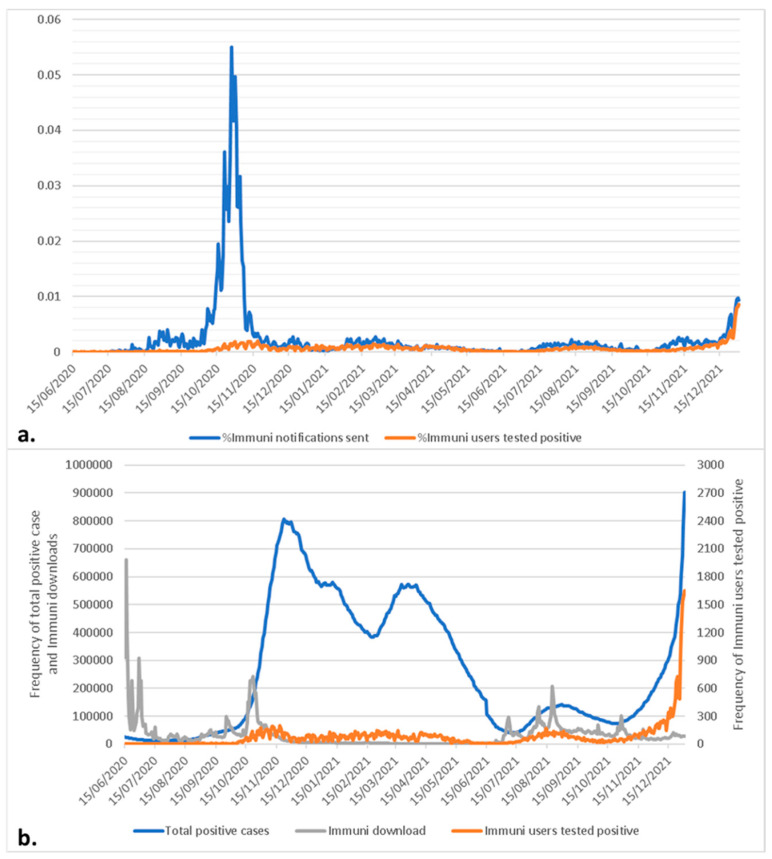
Immuni app data: (**a**) daily percentage of Immuni users tested positive for SARS-CoV-2 and notifications sent by Immuni panel; (**b**) daily frequency of total SARS-CoV-2 positive cases, Immuni downloads (primary axis) and Immuni users tested positive for SARS-CoV-2 (secondary axis) panel.

## Data Availability

All data have been provided by the Ministry of Health and are publicly accessible (Available online: https://opendatadpc.maps.arcgis.com/apps/dashboards/b0c68bce2cce478eaac82fe38d4138b1 (accessed on 31 May 2022)).

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
