# Peer review of "“Immuni” and the National Health System: Lessons Learnt from the COVID-19 Digital Contact Tracing in Italy"

_ijerph, 2022, doi:10.3390/ijerph19127529_

Round 1
Reviewer 1 Report
The Authors present an interesting paper with findings of a contact tracing programs with low penetration.
While DCT has been part of centre stage of all covid prevention efforts, Its adoption, results and efficacy has varied in different regions of the world. I agree with the general outline of the paper and the conclusion drawn, there are some of the concerns and amendments I deem necessary are given below
The inferences the authors have made are in line with the facts listed. But I believe the comparisons have not been extended properly. There are some lessons from NHS England's DCT app and technology which may be relevant for comparison here. Wyman et all in an article in nature 2021 outlined some statistics which interestingly closely resembles those given by canary island spanish study by Rodriguez et all (citation 18). The adoption rates, the secondary attack rates were very similar. This is contrasted by immmuni app, the one whose data that authors have presented, which showed particularly low adoption and attack rate and failure to contribute to national contribution.
- While authors did say that part of reason of non-adoption was nonregularized and semifederalized nature of italian health care system (as mentioned by Armocida et all in lancet) , I believe the article will benefit from further analysis of Immuni vs Spanish/British DCTs in the form of a few lines.
- I think the rest of the manuscript is presented well, and I ll re review th abstract after these changes.
Author Response
Dear Reviewer,
On behalf of my co-authors I would like to thank you for the positive comments on the article and the inputs to further improve the manuscript quality. With this regard, I’m pleased to inform you we have included several findings from the British experience as recommended (in addition, the discussion chapter has been enriched with several other EU experiences other than the Spanish one).
We believe that thanks to your feedback now the article has a much better overview of the phenomenon.
Reviewer 2 Report
Since the early stage of the current pandemic, digital contact tracing (DCT) through mobile phone apps, called “Immuni”, has been introduced to complement manual contact tracing in Italy. Until 31st December 2021, Immuni identified 44,880 COVID-19 cases, which corresponds to less than 1% of total COVID-19 cases reported in Italy in the same period (5,886,411).
The authors reported that: (a) Immuni generated 143,956 notifications. (b)Although the initial download of the Immuni app can represent the interest for a new tool, Immuni has had little adoption across the Italian population and the recent increase in its download is likely to be related to the mandatory Green Pass certification for conducting most daily activities that can be obtained via the application.
They concluded that therefore, Immuni failed as a support tool for the contact tracing system. Other European experiences seem to show similar limitations in the use of DTC leaving open questions about its effectiveness, although in theory contact tracing could allow useful means of “proximity tracking”.
The contribution currently has serious limitations.
With reference to Italy, this theme has already been developed and interpreted, such as in the work https://www.mdpi.com/2227-9032/10/1/67, which the authors have not reported in the bibliography.
These are the problems:
1) An overview of international studies that have faced other Apps for the DCT should be reported in the introduction. In some countries the DCT has ran better. This would help to better understand the reasons for the failure in Italy.
2) It is necessary to clarify the complementary value of their proposal with respect to https://www.mdpi.com/2227-9032/10/1/67
3) At the moment the results are of a few lines and refer to a few data extracted from a public database together with the figures. An extension is needed with further considerations.
4) The discussion is weak and must support the opinion. There is only one international reference, [18] which tells of an international experiment in Spain on the DCT. There is a lack of international works that corroborate the position of the authors.
5) Conclusions are not actually conclusions. They seem a premise for the development of the study "Other European experiences seem to show similar limitations in the use of DTC [21] in the pandemic response, leaving open questions about its implementation in the real- world. As traditional contact tracing remains a crucial component of the COVID-19 re-sponse, mobile apps can offer promise, especially when considering the speed and scale required for tracing to be effective. However, understanding the potential impact of apps as part of a comprehensive integrated approach requires more evaluation of their use in real life and multidisciplinary engagement of several experts (IT, statisticians, public health experts ..).
6) The opinion of the authors, which I struggle to find, must be expressed clearly and concisely and supported by clear and robust evidence.
Author Response
On behalf of my co-authors, I would like to thank the reviewer for the valuable inputs provided.
We have addressed the shortcomings represented by the reviewer as follows:
- The manuscript has been enriched with further inputs from other international experiences (e.g. UK, Germany, Netherlands, Switzerland..).
- Main findings from the referenced publication about Immuni have been added in the manuscript. We would like to clarify that this article has been known by the authors and considered during the conceptualization phase. However, the data proposed by the article are outdated compared to those we analyse hereby and does not consider the Digital Green Pass effect. In addition, the article has tried to address potential drivers leading to the failure of DCT, like digital divide and GDP, based on statistic assumptions (e.g. assuming the digital divide in the country equals to 21%). Therefore, they go beyond the scope of our article. It is notably worth to appreciate that ultimately both the manuscritps came at similar conclusions.
- The result section has been enriched as suggested.
- 6. With the understanding that this manuscript is not a systematic review but a viewpoint on Italian “Immuni”, the conclusions have been further expanded also in reference to point 1 (including experiences from other EU countries as well). The opinion from the authors about the topic have been clarified in the conclusion and it is supported by the evidences hereby represented.
Reviewer 3 Report
This is an informative paper on the adoption and use of apps. Relevant citations of the technology adoption are included in this descriptive study. Although the limitations of the study are noted, more details on the reasons for failing to adopt the technology should be added. For instance, authors should review a variety of theoretical perspectives on the health technology adoption model (Davis) or health belief model (Rosenstock).
Author Response
We thank the reviewer for the valuable inputs provided. Several elements from theoretical perspectives on the health technology adoption model (Davis) and health belief model (Rosenstock) have been added in the introduction and discussion chapters (with the understanding that this is a viewpoint focusing on Italy rather than a general review on the health technology adoption/related models). Thank you again for your comments which ultimately helped us to improve the quality of the work.
Round 2
Reviewer 2 Report
The manuscript improved.
There are not further comments
This manuscript is a resubmission of an earlier submission. The following is a list of the peer review reports and author responses from that submission.